# Natural Intelligence as the Brain of Intelligent Systems

**DOI:** 10.3390/s23052859

**Published:** 2023-03-06

**Authors:** Mahdi Naghshvarianjahromi, Shiva Kumar, Mohammed Jamal Deen

**Affiliations:** Department of Electrical and Computer Engineering, McMaster University, Hamilton, ON L8S 4K1, Canada

**Keywords:** cognitive dynamic system (CDS), software-defined optical communication system (SDOCS), linear Gaussian environment (LGE), nonlinear non-Gaussian environment (NGNLE), perception action cycle (PAC), perception multiple actions cycle (PMAC), cognitive radio, cognitive radar (CR), cyber security, smart e-health, cognitive decision making, self-driving car, cognitive vehicular communications (CVC), fixed transmit waveform (FTW) radar, vehicular radar systems (VRS), cognitive risk control (CRC), coordinated cognitive risk control (C-CRC), mutual interference (MI), smart grid (SG), cognitive control (CC), fore-active radar (FAR), traditional active radar (TAR)

## Abstract

This article discusses the concept and applications of cognitive dynamic systems (CDS), which are a type of intelligent system inspired by the brain. There are two branches of CDS, one for linear and Gaussian environments (LGEs), such as cognitive radio and cognitive radar, and another one for non-Gaussian and nonlinear environments (NGNLEs), such as cyber processing in smart systems. Both branches use the same principle, called the perception action cycle (PAC), to make decisions. The focus of this review is on the applications of CDS, including cognitive radios, cognitive radar, cognitive control, cyber security, self-driving cars, and smart grids for LGEs. For NGNLEs, the article reviews the use of CDS in smart e-healthcare applications and software-defined optical communication systems (SDOCS), such as smart fiber optic links. The results of implementing CDS in these systems are very promising, with improved accuracy, performance, and lower computational costs. For example, CDS implementation in cognitive radars achieved a range estimation error that is as good as 0.47 (m) and a velocity estimation error of 3.30 (m/s), outperforming traditional active radars. Similarly, CDS implementation in smart fiber optic links improved the quality factor by 7 dB and the maximum achievable data rate by 43% compared to those of other mitigation techniques.

## 1. Introduction

Researchers and technology creators and vendors from around the globe are increasingly becoming interested in the Internet of Things (IoT) [1,2,3]. The foundation of Internet of Things (IoT) technologies is the interconnection of various conventional systems and devices, including sensors, actuators, home appliances, televisions, and automobiles with computing devices to enable automatic data transmission across networks. As a result, the Internet of Things creates a network of intelligent systems that can interact with each other and humans [1,2,3]. In recent years, significant developments in computer science, wireless communications, and networking have been made. Many previously unimaginable Internet of Things applications became feasible thanks to small, affordable, low-power sensors, actuators, and electronics components, one example being the intelligent home application for health care in [4]. This application might make our lives more convenient and secure, while also significantly reducing the healthcare costs, especially those for elderly people. In this paper, to design the intelligent system, we focus on the subarea of IoT, i.e., the cyber-physical system (CPS). The smart system’s CPS architecture is depicted in Figure 1.

In Figure 1, the following key components are discussed:**Sensors and Actuators:** environmental sensing refers to a variety of tools and techniques that collect and exchange data about the environment with humans and computers or robots during human–computer interaction (HCI) or human–robot interaction (HRI).**Autonomic computing:** The autonomic decision-making system (ADMS) is in charge of knowledge management, environment situation understanding, intelligent reasoning, and decision-making, including whether or not the user has a disease. ADMS is the name given to this cyber component of the smart system.

The autonomic computing layer (ACL) of a smart system is the main topic of this review. The ADMS can be used to define the ACL [5,6]. The ADMS is tasked with processing data collected by sensors from the environment. Using these data, the ADMS can estimate the environmental conditions, such as whether it is healthy or unhealthy for the human body. Usually, an artificial intelligence (AI) method is used to implement the ADMS. However, in this paper, we review the ADMS for linear and Gaussian environments (LGEs) using a cognitive dynamic system (CDS) concept. We go over several examples of how this functions, such as in cognitive radios and radars (CR), self-driving cars, and smart grids. In addition, we discuss the ADMS’s applications to two NGNLEs—healthcare applications and in fiber optic links—applications that have not yet been thoroughly studied by others. NGNLE, in this context, denotes the absence of Gaussian distributions and nonlinear dependence of outputs on the inputs in nonlinear systems. The perception–action cycle (PAC), language, memory, attention, and the cognitive dynamic system (CDS) are all theories based on the functioning of the human brain [7,8,9]. In order to generate additional rewards, CDS creates internal rewards and takes actions using them [8]. In the majority of applications involving artificial intelligence (AI), this theory has been put forth as a substitute [8,9].

The paper covers various topics related to cognitive dynamic systems (CDS) and their applications, and a summary of the association among the sections/subsections is shown in Figure 2. In Section 1, an introduction and a summary of intelligent systems using CDS as an analogy for the brain is given. In Section 2, we introduce the basic cognition concepts, including the perception–action cycle (PAC), memory, attention, intelligence, and language. Next, in Section 3, we explain why CDS are necessary, discussing machine learning methods such as supervised learning and reinforcement learning. Next, we present the implementation of the principle of cognitions on linear and Gaussian environments (LGEs) using the conventional CDS structure in Section 4. This section also includes various applications, such as the cognitive radio, cognitive radar, cognitive control, and cyber security. In Section 5, we discuss the implementation of the principle of cognition on a non-Gaussian nonlinear environment (NGNLE) with finite memory, including applications in optical communications, software-defined optical communication system (SDOCS), and healthcare. Finally, in Section 6, we present an overall summary of the topics covered in the paper.

## 2. Natural Brain-Inspired Intelligence: Basic Cognition Concepts

Cognitive structures, known as CDS, support adaptive behavior. This adaptive behavior has three main components: (i) the perceptor, which perceives and comprehends the environment; (ii) the executive, which acts cognitively on the environment; (iii) a feedback channel, which transmits internal rewards from the preceptor to the executive. The PACs are understood to be cycles of perception, creation, and transmission of internal rewards through feedback channels, followed by the executive application of actions on the environment based on received internal rewards [10]. This section provides a comprehensive explanation for each cognitive component or aspect.

### 2.1. Perception–Action Cycle (PAC)

A cornerstone of Cognitive science’s theory is the PAC (Figure 3). The PAC is a cybernetics information processing circuit that draws inspiration from neuroscience and the human brain. The PAC aids living things in dynamically adjusting themselves to their environment (e.g., the environment may be a broadcast medium for the acoustic channels, fiber-optic communications, or diagnosing illnesses of a living organism) through goal-directed behaviors or language [8]. In these tasks, CDS perform similar cognitive functions to those of the human brain by processing the measured data from sensors [8]. The fundamental description of the CDS is presented in Figure 3. The function of the PAC as a global feedback loop is shown in this diagram. Both perceptors and the executive are members of the PAC. Additionally, the feedback channel links the perceptor to executive, and the environment completes the PAC.

The following parts are the PAC’s most important components. The first component, a perceiver, processes a collection of observables, or information gleaned from measurements made of the environment. The second component, the perceptor, makes predictions about the environment based on recent and past data using its intelligence and passes those predictions to the executive via a feedback channel. In order for the set of observed values to be updated during subsequent cycles, the third component, the executive one, creates actions in the environment to achieve a specific goal. In the fourth component, each PAC cycle will produce results that will guide subsequent cycles. In the fifth component, the current PAC is directed by derived hypotheses from memory when a particular target is present, an activity is being performed in an environment, or a physical system. In light of this, the CDS revise the hypothesis that will be applied in the following cycle and update the data from the current cycle.

An executive’s actions change the environment and change the status quo. Additional actions are added to this process during round after round until the goal, as determined by system policies, is achieved. The consistent interaction and synchronization of the perception, prediction, actions, and results can be viewed as the PAC. Let us use a woman wanting to drink tea as an example of a goal. She will perceive what it is and what she needs to do with it if she sees the teacup on her desk (sense, observed with her eyes). Even though she has never seen that specific cup before, she is aware of how it differs from a glass of iced water (prediction). This is due to the fact that the brain stores an extracted model for cups and glasses, and she can infer from her prior experiences (extracted models of cups and glasses) that the object she is currently looking at is either a teacup or a glass of water. Additionally, the brain can predict hand movements before the action is carried out (action outcome prediction), and it can also automatically grasp a teacup with the hand (action). Her brain calculates the distance between her hands and the cup after her eyes have perceived it (perception is a result of an action, “picking up a cup”). Until the cup is close to her mouth, this process continues (PAC). Each PAC cycle’s outcomes will influence the cycles that follow. The current PAC is directed by derived hypotheses from the memory when a specific target is present and an activity is being carried out in an environment or physical system. As a result, the CDS modify the hypothesis that will be used in the following cycle and update the data from the current cycle.

### 2.2. Memory

The functional block diagram of the CDS in Figure 4 demonstrates that memory is needed both in the executive (also known as executive memory) and the perceptor parts (also known as perceptual memory) [9]. Executive memory records the choices made or actions taken by the CDS, while perceptual memory enables the perceptor to identify distinctive features of the observable objects and categorize learned features in a statistical sense. The purpose of memories is to learn from the environment, to store learned information, to continuously update the information in response to changes in the environment, and to forecast the effects of choices or actions made by the CDS.

### 2.3. Attention

The perceptor and executive parts, respectively, can use the perceptual and executive attention systems. While the executive part is employed to preserve decision making with the least amount of disruption, the perceptor is tasked with responding to the excessing information from the environment. As seen in Figure 5, a Bayesian model (a statistical representation of an environment) is created before a Bayesian filter is used to estimate that environment. In conventional CDS [9], feature extraction occurs through Bayesian inference, and the executive uses reinforcement learning to reward perception flaws.

### 2.4. Intelligence

The CDS demonstrate an array of intelligence, including problem-solving, perception, reasoning, decision-making, and auto-control tasks. The CDS can make thoughtful decisions that aid it in comprehending unknowable and uncertain circumstances in its environment thanks to the PAC concept and feedback channels.

### 2.5. Language

A network of interconnected CDS can communicate using machine-to-machine (M2M) communications protocols, which opens up a world of opportunities. It is outside the scope of this paper to discuss the creation of a language specifically for CDS, but it is an intriguing idea that has the potential to completely alter how humans and machines communicate.

## 3. Why CDS?

A fascinating and ground-breaking advancement in technology is AI that is powered by machine learning (ML). Machine learning-based AI, as opposed to rule-based AI, which is programmed with predefined rules, can learn from datasets, examples, and experiences. This makes machine learning-based AI extremely potent, but it does have a drawback: in order to produce reliable results, it needs access to huge and trustworthy datasets, examples, and experiences. Because it offers a dependable and effective method of accessing the data and experiences required to create intelligent machines, the CDS has become a well-liked machine learning scheme. This also suggests that training takes longer than it does for rule-based AI in general. In AI that is based on machine learning, a machine learning technique extracts the model from the dataset. The model can then be used to make predictions. The method can also learn to optimize models based on datasets and policies, for example, in the context of a specific job such as maintaining a pre-forward error correction (FEC) bit error rate (BER) under 0.01, while allowing for tolerable modelling complexity.

### 3.1. Machine Learning Methods

In computer science, mathematics, and statistics, machine learning is an interdisciplinary field. In general, there are numerous machine learning techniques, including supervised learning, reinforcement learning, semi-supervised learning, unsupervised learning, and transfer learning. We will only pay attention to the first two major categories, supervised learning and reinforcement learning.

#### 3.1.1. Supervised Learning (SL) Technique

In machine learning, supervised learning (SL) is used for practical tasks such as calculating a patient’s length of stay in the hospital, identifying radar targets, and diagnosing medical conditions. With the aid of the received symbols, it is also possible to estimate the transmitted symbols in transceivers. SL is able to look for patterns in the data. Generally speaking, the SL algorithm can learn how to construct a classifier for estimating the output variable *y* for a particular input variable *x* (see Figure 6a). The SL algorithms can be used to train or extract the mapping function *f* when there is a mapping function *f* (*y* = *f*(*x*)) for the mapping of *x* to *y*. An algorithm that outputs labels {*y*_1_, *y*_2_, *y*_3_, …, *y_m_*} and matches them to a set of data {*x*_1_, *x*_2_, *x*_3_, …, *x_k_*} creates the classifier [12].

Learning by prediction and learning by modeling are the two main divisions of supervised learning. Prediction tasks can use both regression and classification techniques. Regression analysis techniques might be more suitable for predicting continuous output data. For output class prediction, effective classification techniques include support vector machines (SVM), decision trees, and naive Bayes methods. For example, linear regression improves the estimation of a child’s height or the lifespan of an electric vehicle battery. However, decision trees or naive Bayes methods are better for tasks with a discrete number of options, such as binary diagnostic test predictions [12]. These algorithms are capable of determining the decision boundary based on the data and the learning objective. Additionally, machine learning methods such as naive Bayes or Bayesian approaches can be used to extract data probability distributions.

In order to predict the appropriate outputs for new data inputs, the SL algorithm extracts/trains a model on a labelled database (Figure 6a). SL can be used for tasks that require prediction and categorization, such as image recognition and email SPAM filtering.

**Figure 6 sensors-23-02859-f006:**
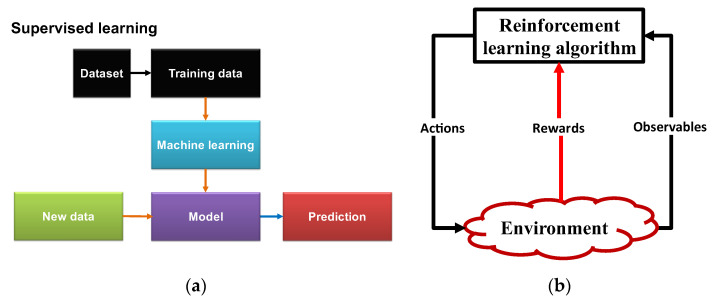
Schematics for two well-known machine learning techniques: (**a**) supervised learning (SL) and (**b**) reinforcement learning (RL) [13].

#### 3.1.2. Reinforcement Learning Method

A schematic representation of reinforcement learning (RL) is shown in Figure 6b. Through trial and error, RL converts a decision-making problem into the interaction of a computer agent with a dynamic environment [14]. When the computer agent searches the state and action spaces, it makes an effort to arrive at the best reward based on feedback from the dynamic environment. For instance, the RL algorithm will attempt to improve the model parameters in healthcare applications by repeatedly simulating the state (user health condition). After performing the action (e.g., the agent receives the feedback reward from the environment (healthy or unhealthy approval status by doctors) by activating or deactivating sensors, adjusting the dosage of medication, and modeling accuracy), the RL algorithm may then settle on a model that produces potential optimal actions [15].

In conclusion, RL learns when it is interacting with a dynamic environment, such as mastering a game or utilizing a system that suggests movies and videos. State and action coexist in real life. Without a database, RL can frequently identify the action that will maximize the reward. However, RL mandates that a model be derived from examples and experiences. RL receives the reward or punishment from the dynamic environment. The executive employs RL to apply cognitive actions to LGEs and NGNLEs in CDS.

### 3.2. Cognitive Dynamic System as Alternative Approach

The executive of the CDS can apply a cognitive action to the dynamic environment using the model that was extracted by the perceptor rather than the conventional RL. The CDS gain independence from the shifting environment, as well as self-awareness, self-consciousness, and internal reward. In other words, the CDS have a conscience and are aware of the action to some extent. For the perceptor to extract the internal reward, the SL algorithm should be applied. The human brain’s imagination serves as inspiration for the prognostication of the results of cognitive actions, similar to a risk assessment before taking any action in the real world. However, in a typical RL, the agent executes the actions blindly in order to gather feedback from the surroundings. The CDS is therefore a better choice than a typical RL is for applications involving intelligent machines.

## 4. Perception–Action Cycle Implementation on Linear and Gaussian Environments

### 4.1. Conventional CDS Structure

The three main subsystems of a conventional CDS, which are inspired by the brain, are a perceptor, an executive, and a global feedback channel that connects the perceptor and the executive, and they can communicate with a linear and Gaussian environment (LGE), such as a cognitive radio [9] (see Figure 7).

#### 4.1.1. Perceptor

A conventional perceptor’s conceptual schematic typically consists of an entropic information processor, a Bayesian generative model, and a Bayesian filter (Kalman filter). For feature extraction in conventional CDS, the Bayesian generative model or Bayesian model is defined. When the observables are available, however, the Bayesian generative model is not necessary [16]. The Bayesian generative model is also an unwise choice due to its high computational cost. As a result, given the environment’s dynamic nature, it is impractical [17]. In the algorithmic implementation in [16,17], the Bayesian generative model was not used. The primary component of a traditional perceptor is typically a Bayesian filter. The state of the observables should be estimated by the Bayesian filter [9,16]. The perceptor frequently makes use of the typical Kalman filter, though, considering that a conventional CDS can only be used in settings with a linear model and Gaussian noise distribution [18]. The entropy of the Kalman filter output can be calculated with the aid of Shannon’s information theory [19,20]. However, the mean and covariance matrix for linear and Gaussian environments (LGE) are all that remain of Shannon’s equation for calculating entropy in typical CDS [16].

#### 4.1.2. Feedback Link of Generic CDS

The executive part receives internal rewards via the feedback channel. When it is using the calculated entropic state of the perceptor as a foundation, the internal reward is determined.

#### 4.1.3. Executive

The planner, actions library, reinforcement learning, and policy are the executive’s most crucial subsystems. Utilizing the cost-to-go function, reinforcement learning’s goal is to maximize the internal rewards received from the perceptor [14]. After filtering by the policy, using the cost-to-go function’s most recent output, the planner should select a set of potential actions from the action library to implement in the environment. The objectives of the planning and reinforcement learning processes that the CDS should enhance are known as cognitive policy in conventional CDS [10].

### 4.2. Brief Literature Survey on Generic CDS

Applications of cognitive radar from the CDS were used in [9,21] to deliver the best target detection and intelligent signal processing. Dynamic spectrum management in wireless communication networks was one of the cognitive radio applications the CDS was used for in [8,21,22]. The theory and applications of the CDS for cognitive control were discussed in [9,10,21]. For risk management in physical systems, the CDS general concept was introduced in [23]. Applications of CDS for risk management have been highlighted for the selection of the vehicle radar transmit waveform [16], the mitigation of cyberattacks in smart grids [17], the detection of cyberattacks in smart grids [24], and the mitigation of vehicle-to-vehicle (V2V) jamming [25]. As the brain of a complicated network, CDS were also employed [26]. An example of a possible CDS block diagram for smart homes can be found in [27]. The authors of [28] discussed the use of CDS in cybersecurity-related applications. In addition to the conceptual block diagram and suggested simulation or experimental results, this review paper discusses the key CDS applications, such as cognitive radio, cognitive radar, self-driving cars, and smart grids (SG).

### 4.3. Cognitive Radio Using CDS

In [9,22,29,30,31,32,33], applications of cognitive radios, such as dynamic spectrum management in wireless communication networks, were used with CDS [32]. The cognitive radio concept using brain-inspired intelligence was introduced in 2005 in [29], which was prior to the introduction of the CDS general concept based on PAC in 2006 and 2007 [22]. There are papers that have introduced cognitive radios without using CDS. In this review paper, we focus only on the cognitive radio that is implemented based on CDS concept.

The cognitive radio was introduced to address the spectrum scarcity for wireless communications. This issue became more critical due to the advances in wireless technology. Due to the exponential growth in global data traffic driven by bandwidth-hungry online services such as high-definition (HD) data streams, the use of wireless systems requiring more bandwidth and an increased demand for a specific spectrum grows significantly. One of the reasons for the spectrum scarcity is the mismanagement of electromagnetic spectrum allocations between the users in an inefficient way [34].

Currently, after gradual adaptation of the new cognitive radio concept, currently, there are two types of wireless communications: the traditional wireless communications system, which does not efficiently manage the radio frequency (RF) spectrum, and cognitive wireless systems that are designed to address the spectrum efficiency management issue in the traditional wireless systems. The unused spectrum sub-bands are called spectrum holes. The cognitive radio interacts with traditional wireless systems dynamically to release these spectrum holes for cognitive radio users. This means that the primary users allow cognitive radio users to maintain communications and perform normal tasks reliably [35].

Interaction between the legacy wireless world and the cognitive wireless world creates a new network of a supply chain and market for the spectrum, where the legacy owners and their customers (primary users) are the sellers and providers of spectrum supply, respectively, and the secondary users (cognitive radios) play the role of the buyers. There are two types of spectrum supply chain networks based on two types of attitudes: (1) allow an open access permission to the spectrum, and (2) allow access to the spectrum based on the market and price. Both types of attitudes play complementary roles because both have different standards, sustainability, and merits. In [33], mathematical models were developed for both types of attitudes and analysis for both transient and steady-state time of cognitive radio. We summarize the principles of cognition by using the cognitive radio as an example:PAC: gain information about the radio environment through PAC, which can be improved from one cycle to another.Dynamic memory: it should be able to predict the outcomes/consequences of actions taken by users of cognitive radio.Attention: It is responsible to moderate between computational cost and achievement of specific goals. It can improve computational resources distributions between cognitive radio users.Intelligence: in the cognitive radio, it is implemented based on principles (1)–(3) that make it possible to perform optimal decision making and data transmission control by cognitive radio users.Language: Typically, it is out of scope in CDS implementation. However, for a specific cognitive radio, it can be defined as the network of cognitive machines. For example, a group of cognitive machines may need to talk each other in a common language to determine coordination in wireless network. In [36], the radio knowledge representation language (RKRL) is proposed as a new idea for cognitive radio networks.

Cognition principles (1), (4), and (5) are similar for the both open access and market-driven attitudes. However, principles (2) and (3) are different based on the selection of the attitude for implementation of a cognitive radio:From the open access perspective, dynamic memory can be built into the radio-scene analyzer (See Figure 8); however, the concept of memory may show itself in dynamic spectrum management [29].From the market-driven perspective, both dynamic memory and attention can be considered as the spectrum broker. Typically, if a human operator performs as the broker, we can assume dynamic memory and attention are implemented naturally. Similarly, the broker can be implemented on a machine using neural computation [18]. Therefore, a machine can be a cognitive spectrum broker that can mediate on behalf of primary users with secondary users. This is similar to “machine-to-machine” (M2M) communication.

In summary, from the perspective of open access, spectrum sensing by the receiver and dynamic spectrum management by the transmitter are the backbone of cognitive radio networks. However, from the market-driven perspective, cognitive spectrum brokers (such as M2M) play a key role in cognitive radio networks. Additionally, when there is an access to a spectrum database scheme, spectrum brokering, spectrum sensing, and spectrum management can be performed by a radio environment map (REM). Since all the information about the network can be stored in the REM, it can improve the cooperation between the primary and secondary users. Some examples of these information are the physical environment, regulation, policies, and primary users’ activity patterns and profiles. Additionally, secondary users can use this database to gain information on the available sub-bands based on their geolocation (i.e., latitude and longitude). Therefore, practically, a cognitive radio network can be created by using non-cognitive wireless devices [37]. For example, in TV white spaces, the radio transceivers do not need be as advanced as typical wireless transceivers do for processing information or sensing the environment, but the database-oriented spectrum access scheme for a TV is similar to the open access attitude. However, spectrum databases should be updated dynamically and frequently [38]. On the other hand, a cognitive radio can be visualized as a radio communication with an internal computer or as a computer that communicates for cyber information processing [29]. As a result, legacy owners and primary users should be paid for resource allocation and computation resources. Cognitive radio should be able to use computational resources through cloud computing based on the demands [38]. Therefore, computational processing should be performed close to the servers so as to have energy efficient radios and reduce the energy consumption of computational tasks. In [39], auction-based competition is suggested for secondary users to access the radio resources, and it can be built on a cloud-based trading engine. Spectrum brokers in a network supply chain can be implemented using this cloud-based trading system.

There are two approaches to have spectrum supply chain network:(1)In the first approach, there is a single tier of spectrum brokers in the network [33]. The broker will decide which user can access to the spectrum.(2)In the second approach, a network has two tiers of spectrum brokers, and these two tiers of brokers can negotiate with each other. One of them negotiates on behalf of the network providers (i.e., legacy owners or primary users), and the other negotiates on behalf of the secondary users (cognitive radio users) [40].

Both approaches have their own advantages. However, the second approach is much more preferred in rural areas [40]. In [41], experimental results based on IEEE 802.11a standard for wireless local area networks reached a satisfactory level, where 48 out of 64 subcarriers were assigned to data transmission, and there are 120 users in the network. Initially, the network had spectrum scarcity issues and users could not transmit the data at the maximum data rate. However, with the passage of time and more iterations of the cognitive radio, the achieved data rate and maximum transmitted power became close to steady state, which was not affected by whether new users joined or left the network [41].

### 4.4. Cognitive Radar Applications

After the successful introduction of cognitive radio using CDS, the concept of cognitive radar in [42]. In [9,21,42,43,44], to provide the best target detection and intelligent signal processing for cognitive radar applications, the CDS was used. In [9,21], the authors developed a cognitive radar system with a cognitive dynamic system (CDS) built within it. The goal was to provide robust information processer that could improve the tracking target process and be intelligent enough to choose a right radar pulse depending on the target situations [9,21,42,43,44].

The part of the human brain responsible for visualizing is one of the most powerful parallel information processors machines ever known. The visual brain can do many tasks such as accurate and reliable target detection/recognition and tracking. The speed of visual brain is far faster than that of any traditional radar system known today [42]. There are some similarities between the human visual brain and traditional radars. For example, they both visualize the world (environment) or tracking objects. Radar is an active sensor which detects the reflection of the transmitted signal from objects and environment. Then, it builds a picture of the environment through a decision-making process based on the received signal, which is the reflection of the transmitted signal. Therefore, the radar should perform actions to improve the reflected signal from the target to the radar, and the radar should perceive the return signal as an active sensor [44]. Natural brain-inspired intelligence can be used to improve the information processing of traditional radar. This innovative idea is known as cognitive radar (CR) that relies on the principles of perception action cycle (PAC). The CR was inspired from bat echolocating mechanism in [45] and, then, the PAC concept of the visual brain in [8,9]. CR is the enhanced version of fore-active radar (FAR) proposed in [46]. In FAR, an offline link between the receiver and transmitter is added, and the traditional active radar (TAR) is converted to closed loop control system, FAR. The difference between three types of radar is as follows:(1)TAR operates in a feed-forward manner; TAR can make use of adaptive filtering such as Kalman filtering. Therefore, it can estimate the iterative state in the TAR receiver [47]. TAR may adaptive beamforming to find the target using the transmitted waveform [48].(2)FAR can be distinguished from TAR by a feedback link connecting the receiver to the transmitter. FAR is also known as fully adaptive radar that has global feedback and environment. FAR can perform adaptive filtering by the receiver and adaptive bream forming by the transmitter in a similar way to that of TAR [9]. In terms of the theory of control, it is well known that the feedback used in FAR is more intelligent than TAR is. However, FAR/fully adaptive radar has limited intelligence and it is just a first step towards cognition, whereas radar should perform practical use of transmit waveform selection in transmitter [47,49,50] or resource allocation [51].(3)Cognitive radar (CR) has extra capabilities that makes it different from TAR and FAR: CR can develop rules and behaviors with self-conscience and self-organizing using the PAC concept and gain experience by interacting continuously with the environment [42].

The CR block diagram using the PAC concept is shown in Figure 9. The waveform adaptation is performed in the PAC. Using waveform adaptation, CR can gain control over the sensing and observation process. That means the PAC guarantees estimation by sensing and control in the CR [52]. As a result, CR has the ability to mitigate the effects of disturbance and anomalies such as clutter. The input of the preceptor in the receiver is the return transmitted signal from an unknown target and should fulfill two goals: (1) reliable target detection and (2) tracking the target behavior during the process. Target tracking can be defined as a Bayesian framework state estimation problem or Bayesian filtering [9].

The traditional approach to target tracking uses pair of equations of a state-space model: (1) the first equation shows the evolution of the state with time in the presence of noise in the radar system and modules, (2) and the second equation shows the dependency of the measured signal at the receiver to the environment noise that corrupts the received signal.

In some radar systems, the model of state space is nonlinear, but the noise is Gaussian. Therefore, the approximation of optimal Bayesian filtering is achieved in some tracking scenarios [9], where the cubature Kalman filter (CKF) [50] estimates the state space for these scenarios. In other words, the approximation of the Bayesian filter (CKF) is used to perceive the environment, and it is performed by the receiver.

Linking the receiver to the transmitter is required in the PAC. In a monostatic radar, it is straightforward because the receiver and the transmitter are at the same place. Therefore, the receiver provides the computation of feedback information to the transmitter for performing the action in the environment.

Mathematically, state estimation can be considered as the perception. Additionally, mathematical representation of the feedback information can be achieved using the state estimation error vector [9]. CR executive in the transmitter can indirectly control the receiver using an antenna by illumination of the environment. In [9], for the optimal control, the Bellman’s dynamics programming (BDP) is used as an optimization technique to perform reinforcement learning (RL). However, BDP cannot be used in practical CR systems due to issues such as higher dimensional action space, measurement space, or combination of action space with measurement space. Therefore, in [10], an approximation of BDP is used to mitigate the complexity cost and the curse of dimensionality problem.

The task of the perceptual multiscale memory (see Figure 9) is to select the important extracted features of the transmitted signal, and that of executive multiscale memory is to reduce number of actions in actions space (e.g., transmitted waveforms library) for the BDP approximate algorithm. In Figure 9, working memory function is a neural network of the heterogeneous associative kind [18,51,52,53,54,55]. For example, important features in the model library grid should be matched with the selected transmitted waveform at the transmitter. If an error happens, that means wrong features are selected in the multiscale perceptual library. The transmitted waveform working memory will fix this mistake in the next cycle by sending specific action from the action space chosen by multiscale executive memory.

In summary, the PAC in CR is applied in four subsystems: (1) the perception of environment using approximate Bayesian filter/CKF in the receiver, (2) the feedback link between the receiver and the transmitter, (3) approximate BDP controls the transmitter waveform inside the transmitter and, (4) the state-space model of the CR environment. The PAC can be viewed as a closed-loop feedback control system [10] (see Figure 9). The PAC is responsible for the generation of/gaining information from the CR environment by processing the received signal that is the reflection of transmitted signal, and the amount of gained information should be increased from cycle to cycle.

Table 1 shows the root mean square error (RMSE) for the range and range rate for TAR, FAR, cognitive radar with on layer memory (CR1), CR with multiscale memory (CRm), and cognitive radar with full attention and memory (CRa) [49]. According to Table 1, all types of CRs perform better than TAR and FAR do (uses only feedback). CRm performs better than CR1 does. CRa performs better than CR1 does, but the CRa performance is slightly worse than that of CRm. The reason is that in CRa, the computational cost is reduced at the cost of the performance reduction. However, the performance improvements of CR were the result of the additional computational cost of $O({\left(N_{g}\right)}^{L})$, where the $N_{g}$ is the size of the waveform library and *L* is the length of horizon of dynamic programming [49].

### 4.5. Cognitive Control

Cognitive control (CC) is a paradigm that was first introduced in 2012 by Haykin et al. [9] and is additive in nature, rather than a replacement system design paradigm. The details of the cognitive control principles using CDS are discussed in [10,18,56,57]. The cognitive controller in a cognitive dynamic system (CDS) is inspired by two fundamental aspects of the human brain, namely, learning and planning.

Learning: CC learns based on two basic ideas:To bypass the imperfect-state information issues, the CC can use the entropic state of the perceptor [56,58].CC can use PAC as the first principle of cognition [8,9].Planning: CC can plan a process that is inspired by the prefrontal cortex in the brain [58,59]. Particularly, the cognitive controller in the executive is reciprocally connected to the cognitive preceptor at the perceptor. This reciprocal linking can use feedback information sent by perceptor to the controller, and the perceptor can receive feedforward information from the controller in the executive part (Figure 10). In [10], CC is studied in terms of linearity, convergence, and optimality, which are three fundamental properties of the CC learning algorithm, and they are described as follows:
The linear law of computational complexity can be calculated based on the number of actions taken on the environment, based on the learning algorithm.Since the RL algorithm in the executive uses a specific case of the Bellman’s dynamic programming, the convergence and optimality of the learning algorithm can be automatically approved [10].

The CC algorithm in [10] is validated experimentally with a cognitive tracking radar benchmark. Particularly, the CC algorithm in [10] used for planning was compared with tracking radars without CC, CC without planning, and RL (here, Q-learning). Although CC is computationally more expensive than the other algorithms are, CC outperform significantly radar with a fixed waveform shape, RL, and CC without planning.

### 4.6. Cyber Security

In [28], the use of CDS for cybersecurity applications was discussed. In [28], it is conceptually shown that the AI cannot address cyber security issues, whereas the CDS can address this issue in an active way. Cybersecurity is implemented in the CDS executive by using the cognitive predictive action for the environment [28].

Historically, it is helpful to know how the AI started in the 1950s. Alan Turing, a mathematician, wrote a paper titled “Computing Machinery and Intelligence” [60]. Then, this mathematical field produced AI. AI is often considered to have the potential to impact the practice of mathematical modelling [28]. Regarding some important aspects of mathematics as we know them today, we may mention the following issues that are important: (i) unsupervised learning, (ii) supervised/ deep learning, (iii) deep reinforcement learning, and (iv) federate learning [18]. In [28], it was stated that none of these four aspects of mathematics (typically known as AI) has been able to solve the cybersecurity problem. The authors of [28] provide concepts and statement that indicate that the cyber security problem should be solved using the cognitive control aspects implemented in executive CDS.

#### 4.6.1. Cyberattack in the Smart Grid

In [24], CDS are presented as a supervisor for the smart grid (SG) network, as shown in Figure 11. In Figure 11, the perceptor receives measurements from the smart grid and evaluates them to generate a posteriori for the entropic information processor to create an internal reward (entropic state). This entropic state is then sent to the executive to assess whether there is a fake data injection (FDI) or not. If there is, the policies and planner take necessary actions to mitigate it. CDS are used to detect [24] and mitigate [17] the cyberattack in the smart grid. The CDS can recognize the incorrect measured data by sensors. So, CDS can play a key role in managing the SG because the incorrect measured data can damage the hardware and the gird, and even domino cascade effects can disrupt the whole grid performance, whereas a typical control system may not be able to mitigate this problem [16].

In [24], the direct current (DC) state estimator is considered as the environment in which the CDS should testify the measurements from the SG. Here, the generative model classifies measured observables from the environment based on the cumulative sum method [24]. Other details are similar to generic CDS for LGE that have already been discussed previously. The cognitive control function in the executive assigns optimal weights to each sensor. Therefore, CDS can provide the optimal DC state estimation by reducing the measurement errors. Similarly, the optimal DC state estimation can be achieved in presence of a cyberattack, where may try to inject wrong data to the network.

Two different experiments were conducted based on the algorithms presented in [16]. The CDS in the first experiment successfully detected the bad data and corrected them. In the second experiment, it is found that the entropic state in the CDS can be used as the metric for internal rewards for cyberattack detection. Thus, Ref. [16] demonstrated that CDS can detect cyberattacks where the attackers try to use fake data injection (FDI), and entropic state can be used as a good metric for the FDI. However, due to high computational costs, the framework in [24] cannot be applied to a real large-scale smart grid system. Using machine learning techniques such as neural networks may reduce the computational cost [24].

In [17], cognitive risk control (CRC) was added to CDS to bring the risk under control in uncertain situations such as cyberattacks, in a SG. Similar to [24], it was demonstrated in [17], that the entropic state of CDS can identify FDI attacks, and then the executive part can control the FDI by taking cognitive action. As a result, the CDS oversees the SG. The results in [17] showed that this system has great potential for future SG systems, where the computational cost concerns for the algorithms can be addressed by more powerful processors, despite the fact that the computational cost is too high to implement in practical SG.

#### 4.6.2. Self-Driving Cars as a Combination of Cognitive Radar, Cognitive Radio, Cognitive Risk Control, and Cyber Security

The implementation of CDS for CR and cognitive radios opened the door for more applications of CDS for self-driving cars, which use radar and communications systems. CDS have found applications for vehicular radar transmit waveform selection in [16], for risk control, cyber security and anti-vehicle-to-vehicle (V2V) jamming in [25], and a combination of anti-V2V jamming for risk control, cognitive radios, and CR implemented for self-driving cars in [60]. This is depicted in Figure 12, which illustrates the cognitive mediator’s role in assigning a specific frequency and bandwidth to CR and cognitive vehicular communications. CR can be leveraged to increase the environmental awareness of nearby vehicles’ distances and speeds, while cognitive vehicular communications enable communication between cars to minimize the risk.

There are potential risks and hazards posed to autonomous vehicles in the presence of uncertainty, jams, and hacking. Therefore, the task of CDS is to improve the performance of vehicular radar systems (VRS) and their V2V communication systems in such dangerous cases. The authors of [16,25,60] present the architectural structure of the cognitive risk control (CRC). In [23,61], the general idea of CRC using CDS for risk control in physical systems was introduced, and specific applications of CDS for risk control were presented, such as self-driving cars and smart grids.

In vehicular radar systems, CRC can intelligently and adaptively select the transmit waveform. Details of coordinated cognitive risk control (C-CRC) implementation, dynamics, and the modeling of self-driving cars in different scenarios are provided in [62]. A mm wave radar with a carrier frequency of 77 GHz and with linear modulation is used to test the performance of CRC for VRS and V2V communications under various risky conditions. In [62], the simulations were performed for jamming the VRS for C-CRC in [62], C-CRC in the presence of mutual interference (MI) (C-CRC MI) [62], CRC only in [16], fixed transmit waveform (FTW) radar (Typical radar), and Q-learning (Reinforcement learning) in [63]. The simulations were performed for four different cases [62], and each case represents a practical scenario that a self-driving car may face in a real environment. Case one is the ideal case scenario, i.e., no disruption is caused by the environmental actions or jamming/hacking attacks. When situation two occurs, the jammer has the option to abruptly alter its movement without using jamming technology. In scenario three, the jammer motion is consistent and fluid. The real V2V link, however, may begin to be attacked by the jammers. The worst case scenario is regarded as case four, in which the jammer may suddenly change motion, while simultaneously attacking the V2V system.

For VRS, the simulation results for all the cases showed that CRC only, C-CRC, and C-CRC MI outperform the FTW and Q-learning techniques by up to 70% [60,62]. Additionally, C-CRC and C-CRC MI perform 10–50% better than CRC does only depending on these scenarios. As expected, C-CRC interference performance is slightly better than the CRC MI. The reason is that C-CRC MI should face mutual interference between the vehicles as an additional source of disturbance.

In [60,61], similar simulations were performed for cognitive vehicular communications (CVC), in which comparisons were performed between the fixed transmitted power (fixed), random transmit power selection, CRC only, C-CRC, and C-CRC MI algorithms for cases 1–4. In all the scenarios, the C-CRC algorithm performed better than the rest of algorithms did. Additionally, in the presence of the MI, the received signal-to-interference-and-noise ratio (SINR) decreases, and the results are worse for C-CRC MI and similar to those of VRS waveform selection. The results of C-CRC MI are only slightly better than those of the type 2 CRC only, even in the presence of MI.

## 5. PAC Implementation on NGNLE with Finite Memory

Figure 13 depicts the conceptual implementation of the CDS for the NGNLE. The CDS for NGNLE applications can be thought of as an improved AI because they combine SL and RL, the two main approaches of ML-based AI. The model is extracted by the CDS perceptor using the SL algorithm. The extracted model can be used by the perceptor to generate internal reward and predict the outcome of the dynamic environment [9,13] (see Figure 13 also). NGNLE with finite memory is a dynamic environment (e.g., fiber optic connection). The perceptor informs the executive of the internal reward through the feedback channel. The internal reward in the current PAC is used by RL in the executive to determine a course of action that will either lower or raise the internal reward for the upcoming PAC. However, before applying actions to a real environment, the CDS can forecast the results of the actions using a virtual NGNLE to ensure that the action can optimize the internal reward within a range pre-defined by policy requirements.

### 5.1. Evolution of New CDS Designed for NGNLE

The traditional CDS in [2] cannot be applied in nonlinear and non-Gaussian environments (NGNLEs), such as long-haul fiber optic links and medical applications, according to [13]. Since the conventional CDS were only meant to be used in linear and Gaussian environments (LGEs) [9,10,16,17,18,19,20,21,22,23,24,25], LGE here means that the environment’s outputs have a Gaussian distribution and depend linearly on the inputs. The conventional CDS perceptor employs the Kalman filter, which can only be applied to LGEs and cannot be applied to NGNLEs. Two additional issues are the hardware implementation of the Kalman filter and its computational complexity. The conventional CDS perceptor is unable to extract a model with finite memory, which is a requirement for CDS implementation on an NGNLE. The equations of conventional CDS can also be simplified for LGEs. For example, it is possible to transform the equations for the entropic state and the entropic processor into mean and covariance matrices, respectively [16]. Due to the intricate equations required for an NGNLE, the conventional CDS is impractical for use in complex environments, such as long-haul nonlinear fiber optic links or healthcare applications. A CDS for NGNLE with simple, quick PAC algorithms was developed in [7,11,13] to obtain around these limitations.

CDS are redesigned in [9,10,16,17,18,19,20,21,22,23,24,25] using the PAC concept so that they can be used with NGNLEs. A computationally effective performance is desired for the redesigned CDS. Therefore, for extracting the posteriori information, the authors of [9,10,16,17,18,19,20,21,22,23,24,25] substitute subsystems such as the entropic state processor or Kalman filter with assurance factor and supervised learning, respectively. These new subsystems have a lower computational cost than the equivalent traditional CDS subsystems do, allowing the CDS to be applied to NGNLE. Additionally, some subsystems have been added to the CDS that are inspired by human imagination and predict action outcomes before applying actions to real NGNLE, such as the executive creating a virtual NGNLE for the purpose of doing so.

#### 5.1.1. NGNLE Applications and Examples

The majority of the data gathered or measured from social sciences, education, and health conditions are not normally distributed [63,64]. A few examples of non-Gaussian distribution in engineering, social sciences, education, and health conditions, respectively, include the modeling of prostate cancer [64,65,66], psychometrics [67], labor income [68], and nonlinear fiber optic links [11,69]. In Section 5.1 and onwards, we present the CDS subsystem for NGNLE based on [13] (see Figure 13).

#### 5.1.2. Perceptor for NGNLE

The three main subsystems that make up the preceptor of [13] are (i) supervised learning (SL) for extracting the posterior, (ii) cognitive decision-making based on the maximum of posteriori (MAP) rule, and (iii) internal reward calculation using the assurance factor concept in the perceptor of the CDS.

(i)Supervised learning (SL): For the purpose of extracting the posteriori of NGNLE, the perceptor builds an automatic decision tree or a forest of trees [13]. An approach that is frequently used in SL is the decision tree approach. Depending on the application, the most recent CDS version can produce decision trees in the perceptor in an adaptive manner [13].(ii)MAP rule-based cognitive decision-making: The CDS uses a MAP rule for cognitive decision making in versions 1 through 6 [7,11,13], extracting posteriori data from the SL. The MAP rule’s finer points are covered in [13].(iii)Utilizing the assurance factor concept, internal reward calculation: The internal reward is modeled after fuzzy human decision making with a lower computational cost, and is especially useful for complex NGNLEs, such as those used in healthcare applications. Fuzzy logic in this context simply means that the logic values of variables can be any real number between 0 and 1 [69,70,71,72,73]. Fuzzy logic can be viewed as a decision assurance. For example, we can make the wrong decision when the assurance is less than one.

#### 5.1.3. Feedback Channel (Main Feedback and Internal Commands)

There are three different kinds of feedback channels: the global feedback channel, the internal feedback channel (from the executive part to the perceptor), and the internal commands link. The internal reward is sent to the executive through the global feedback channel, such as in a traditional CDS. The extracted model and all other necessary data from NGNLEs or current observables can be sent by the perceptor to the executive through the internal feedback channel. The executive can give internal instructions to the perceptor for adaptive modeling configuration, such as the level of decision trees or the accuracy of discretization. In other words, the internal commands are inspired by the machine learning engineers who play with modeling parameters of different types of machine learning techniques to achieve the desired results within the acceptable computational cost for training, decision making, and estimation of NGNLE situations.

#### 5.1.4. Executive Part of CDS for NGNLE

The executive part is a crucial component of the CDS in [13], similar to the conventional CDS [9]. The executive part for NGNLE, however, uses streamlined RL techniques with the lowest possible computational cost, unlike the traditional CDS. Executives for NGNLEs are in charge of enhancing both the decision-making accuracy and the speed at which predetermined objectives, such as a diagnosis error less than 10%, are attained [13]. This can be achieved by acting on the NGNLE or by giving the perceptor internal commands. The executive uses unconventional reasoning, i.e., by using the internal reward and altering the decision tree level, the CDS can invalidate the prior decision, while gaining more evidence. The proposed NGNLE executive consists of three parts: (i) a planner (which also includes a library of actions), (ii) a policy, and (iii) learning based on predictions of the results of hypothetical/virtual actions.

(i)Planner and library of actions

The planner pulls out the collection of anticipated actions that are already saved in the CDS actions library. The actions library, in this case, is a collection of every action that CDS are capable of performing within the action space. Additionally, the planner chooses the initial action from the first PAC using pre-determined actions or randomly from a pool of actions in the actions library. The type of actions in action space can be environmental actions or internal commands. Environmental actions can be applied to NGNLE. In addition, the planner updates the actions type and sends an internal command to the perceptor for updating modeling configuration.

(ii)Policy in the executive

The desired outcomes that the CDS should accomplish with the PAC are established by the policy. The objective could involve making a trade-off between the computational expense and the desired CDM accuracy. This objective could, for instance, be to provide a bit error rate (BER) of less than 4.7 × 10^−3^ for the highest data rate or to diagnose a patient’s health state with a decision-making error of less than 10%.

(iii)Learning through the use of predicted virtual NGNLE action outcomes

The goal of reinforcement learning, such as that of conventional CDS, is to maximize the inbound internal rewards computed in the perceptor and received through the global feedback channel by the cost-to-go function [14]. However, the NGNLE executive can act in the virtual environment and maximize the internal rewards without having to do so on the actual NGNLE. For instance, the CDS might require 20 PACs to accomplish the predetermined goal in the absence of action prediction results. However, the CDS can achieve a predetermined goal in nine PACs when using a virtual environment to predict action outcomes.

### 5.2. State-of-the-Art CDS Versions Are Now Available for NGNLE

In contrast to the traditional CDS in [9], six CDS versions have been published for NGNLE, each with PAC algorithms that are straightforward, fast, and simple [7,11,13,71,74,75,76,77,78,79]. For these versions, the authors of [7,11,13,71,74,75,76,77,78,79] present the CDS concept, design, and algorithms. In next sections, we go over these six versions in greater detail. The following is a brief description of these versions’ development (see Table 2).

CDS v1: An example of a NGNLE is a long-haul fiber optic link, which uses the CDS v1 with a basic executive [11]. The CDS v1 a simple executive, however, it is unable to foresee the results of actions before putting them into play. Additionally, the modeling settings of the perceptor cannot be controlled by the simple executive.CDS v2: In ADMS and for cognitive decision making on an NGNLE, the CDS v2 is used as an alternative to AI [71]. Increased or decreased decision tree levels can be controlled by internal commands in CDS v2 to alter the perceptor’s modeling configuration. This variant of CDS can therefore simulate an NGNLE with finite memory. Using a virtual NGNLE, the executive, however, is unable to foresee the results of the actions.CDS v3: As a general idea, design, and set of algorithms for the CDS for the CDM in NGNLE, CDS v3 is presented. The advanced executive is utilized by CDS v3 [7]. Using a virtual NGNLE, the advanced executive can forecast the results of several actions before implementing one in the environment. Additionally, the advanced executive can use internal commands to modify the perceptor’s modeling configuration. An NGNLE with finite memory, however, cannot be modeled by the CDS v3 perceptor.CDS v4: As an improved general purpose algorithm of the CDS for a CDM system in an NGNLE with finite memory, the perceptor and executive of CDS v3 are upgraded [13]. Therefore, CDS v4 could be seen as a more generalized version of CDS v3, and the perceptor could extract the NGNLE model with finite memory. CDS v4 employs a sophisticated executive that can foresee the results of numerous actions before executing one on an environment with limited memory. The advanced executive can also alter the perceptor’s modeling settings via internal commands and alter the decision tree’s level to alter the focus level.CDS v5: The inherent weakness of ML-based AI in comparison to rule-based AI is the reliance of ML-based approaches on trustworthy datasets [74]. As a result, ML-based AI performs significantly worse when a dataset is defective. In this context, a defective dataset refers to one that either lacks sufficient training patterns, has inadequately labeled training patterns, or has both problems. Human errors or a covert cyberattack are both potential causes of a flawed dataset. To lessen the impact of a flawed dataset and produce accurate results, CDS v5 employs conflict-of-opinion (CoO) decision making [75]. In contrast to the other CDS versions (CDS v1–v4, v6), CDM based on CoO are located in the executive rather than the perceptor. The perception multiple actions cycle (PMAC) concept is a generalization of the PAC concept in CDS version 5. The NGNLE model can be extracted by the CDS v5 perceptor as a forest of decision trees.CDS v6: The perceptor is crucial for implementing the CDS for software-defined optical communications systems (SDOCS) [79]. The posterior of SDOCS should be easily extracted by the CDS perceptor [79]. The authors of [7,11,13,75,76] apply the CDS to fiber optic systems based on orthogonal frequency division multiplexing (OFDM) and compute the posterior using a Bayesian equation. The authors of [79] introduced CDS v6, in which the posterior is directly extracted during the training phase and decision making with no real computational cost associated with multiplication. CDS v6 is implemented for 15 Tb/s optical time division multiplexing (OTDM) systems, unlike the CDS v1, 3, and 4.

**Table 2 sensors-23-02859-t002:** Six versions of CDS have evolved.

CDS Version	Virtual Actions	Internal Commands	Modeling with Memory	Cognitive Decision Making	NGNLE for Proof of Concept Case Study	Direct Posterior Extraction
V1: Simple CDS [11]	🗴	🗴	🗴	MAP ^1^ rule	OFDM long haul fiber optic link	🗴
V2: ADMS [7]	🗴	✓	✓		Diagnostic test (Health)	🗴
V3: Advanced CDS [72]	✓	✓	🗴	MAP rule	OFDM long haul fiber optic link	🗴
V4: Advanced CDS with focus level [13]	✓	✓	✓	MAP rule	OFDM long haul fiber optic link	🗴
V5, ADMS with non-monotonic reasoning ^2^ [75]	🗴	✓	✓	CoO	Health screening	✓
V6: Advanced CDS with direct posterior extraction [79]	✓	✓	✓	MAP rule	15 Tb/s OTDM fiber optic link	✓

^1^ MAP rule: Maximum posterior rule. ^2^ Non-monotonic reasoning: Adding additional information or knowledge can invalidate a decision.

#### 5.2.1. CDS for Optical Communications

By sending lightwave signals through a fiber-optic link, fiber-optic communication systems can transmit information from one location to another [73,74]. An optical transmitter, an optical fiber, and an optical receiver are the three main subsystems of a typical fiber-optic link. An electrical signal is changed into an optical signal by the fiber optic communication system’s transmitter using an optical modulator. The fiber optic communication system’s receiver transforms the optical signal into an electrical signal after it has passed through the optical fiber. The main benefit of optical fiber communication systems is the enormous bandwidth (10 THz) that is available, making it possible for high data rates (Tb/s) to be attained. This is mainly due to the fact that the optical communication system’s carrier frequency is a high-frequency laser with a center frequency of about 200 THz. This center frequency is much higher than the microwave or mm-wave systems’ (300 MHz–300 GHz) center frequency.

#### 5.2.2. CDS as the Brain of SDOCS

A software-defined optical communication system (SDOCS) is an illustration of a non-Gaussian nonlinear environment (NGNLE) [79]. In [7,11,13,71,74,75,76,77,78,79], a new CDS was introduced for SDOCS applications for the optical network physical layer, and the CDS was redesigned for NGNLE applications based on the PAC principle. If different connection lengths, waiting times, or bandwidth requirements are needed, the CDS on SDOCS should be able to accommodate them. The transceivers in SDOCS can be optimized in terms of performance and are programmable and reconfigurable (see Figure 14). The concept of smart SDOCS upgraded by CDS as the brain is depicted in Figure 14. CDS have the ability to support numerous client services by selecting various overhead coding schemes, adaptive modulation formats, and necessary digital signal processing (DSP) in transceivers. Furthermore, CDS are capable of tuning a different number of optical subcarriers in SDOCS.

Similar to software-defined networks (SDNs), the SDOCS can support flexible optical channel adaptation by adjusting the transceiver’s parameters. By updating the fiber optic systems with CDS, which can optimize the optical link parameters, the SDOCS objectives can be met. Figure 14 depicts the idea of a smart SDOCS that is upgraded by a CDS acting as the brain. Numerous client services are supported by the CDS. The required digital signal processing (DSP) in transceivers, as well as various overhead coding schemes, and adaptive modulation formats can be chosen. Additionally, the CDS can adjust the SDOCS’s optical subcarrier count in different ways [7,11,13,75,76,77,78,79].

**Figure 14 sensors-23-02859-f014:**
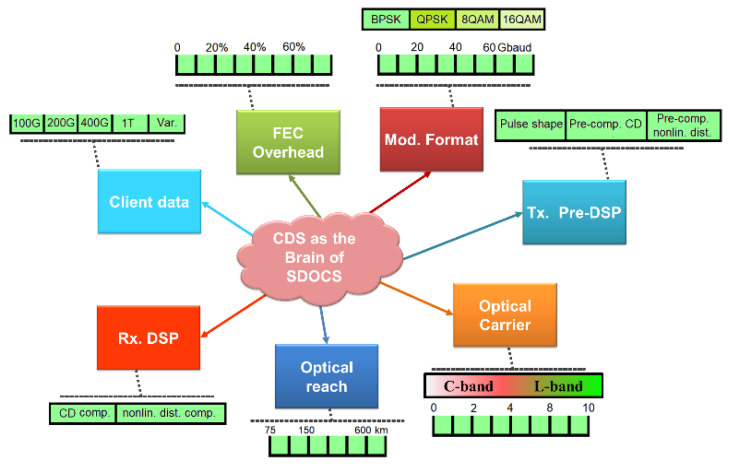
The software defined optical communications system (SDOCS) relies on CDS as its brain.

The simple CDS, also known as CDS v1 for NGNLE, is presented in [11,76]. It is demonstrated that CDS v1 can make fiber optic links more efficient and intelligent (even when there are disturbances present). CDS v3 or CDS with the action outcome prediction for an NGNLE is presented in [7,75]. In [7], it is shown how the internal commands for the adaptive modeling configuration and the prediction of actions’ results lead to an improvement in data rate and speed up the CDS’ ability to accomplish predefined goals. CDS v4 is presented for NGNLE with finite memory in [13]. For extracting adaptive modeling of NGNLE with finite memory, CDS v4’s executive and perceptor have both been upgraded. Implementing CDS v4 for long-haul fiber-optic links demonstrates that adaptive modeling with finite memory can significantly increase the data rates for long-haul fiber-optic links compared to those of earlier versions of CDS. Such as CDS v3, CDS v4 has the ability to use the action outcome prediction applied to a virtual NGNLE in the executive. The ability to predict how actions will turn out makes CDS v4 faster than the earlier iterations are at achieving set objectives [13].

As stated in Section 5.2, CDS v1, v3, and v4 were implemented for long-haul fiber-optic links as an example of an NGNLE. This is a typical function of the CDS in a fiber optic communication system; it can always keep the decision-making error below the system threshold. Furthermore, unlike digital backpropagation (DBP), the CDS does not need the fiber-optic system parameters, such as the quantity of fiber spans, the length of each fiber span, its dispersion, and its nonlinear coefficients. A statistical model of a fiber-optic channel can be obtained through intelligent perception processing by the CDS. Additionally, the CDS is able to learn and identify optical network disturbances such as changes in the data rate, fiber length, or power fluctuations. A broken noisy amplifier can cause a significant disturbance in the system, but CDS v1 can handle it and improves the Q-factor by 2.74 dB, while increasing the data rate by 12.5 percent [11]. Using quick algorithms, CDS v3 can increase the data rate efficiency by 23.3 percent and Q-factor by 3.5 dB [7]. Additionally, the CDS v3 accomplishes the predetermined goal faster (in 8 PACs), whereas the CDS needs 12 PACs if the action outcome prediction is not made. CDS v4 can increase the data rate efficiency by 43% and Q-factor by 7 dB [13]. In addition, CDS v4 can complete the task in nine PACs. A CDS, however, calls for 20 PACs without the executive’s prediction of action outcome.

The perceptor is essential for fulfilling the SDOCS requirements in order to implement CDS for the SDOCS [78]. Even during the training phase, the CDS perceptor should be able to efficiently extract the posterior of SDOCS. The CDS is used with fiber optic systems based on orthogonal frequency division multiplexing (OFDM) in [7,11,13,75,76]. The Q-factor of a single channel OFDM system was improved from 3 to 7 dB using CDS numerical simulation results in [7,11,13,75,76], along with an increase in data rate from 12.5 to 43%. Internal rewards and a report on the state of SDOCS are sent to executive by the perceptor. Executives implement actions such as adjusting the modulation format (see Figure 14) or altering the data rate based on internal rewards. The executive also takes steps to realize SDOCS’s desired objectives, such as balancing the maximum feasible data rate with the maximum practicable transmission distance. Additionally, as a side benefit, the CDS can reasonably reduce optical nonlinearity using the extracted posterior and CDM.

A technique for direct posterior extraction Is used by CDS in [78]. The perceptor in [79] requires less computational effort than those in [7,11,13,75,76] do, where the posterior is determined by the Bayesian equation using the extracted model and data. Additionally, the perceptor in [79] can make up for fiber channel nonlinear impairments, just such as the perceptors in [7,11,13,76,77]. The application of machine learning for nonlinear compensation of ultra-high-speed optical pulse transmission, such as an optical time division multiplexing (OTDM) system, is a major driver for using CDS. This is because it is very challenging to achieve with other DSP techniques, such as DBP. The received data of all the OTDM tributaries would be needed for precise DBP-based nonlinear compensation in OTDM transmission (128 in [78]), which is practically impossible. The receiver does not have access to numerous WDM channels in optical networks, which is actually the reason DBP is hard to apply to WDM data. Uncertainty about the data of other OTDM tributaries is avoided thanks to CDS v6’s ability to compensate for nonlinearity using only a single tributary’s information. Additionally, the CDSv6 perceptor does not necessitate prior knowledge of the fiber channel parameters, such as span lengths, fiber dispersion, loss, and nonlinear coefficients of each span.

Without relying on the assumption of Gaussian noise, the CDSv6 perceptor directly extracts the optical link posterior (main task). In [78], it is experimentally shown that CDS v6 can reduce unexpected optical link distortions such as digital clock recovery intolerance (QoT control and robustness requirement for SDOCS, a tertiary benefit) in addition to fiber nonlinear impairments (secondary benefit). The received signal passes through the perceptor of the CDS during a single-channel 15.3 Tb/s [79] polarization-multiplexed 64 QAM OTDM transmission experiment over a 150 km fiber (see Figure 15). In the beginning, the posterior could be successfully and efficiently extracted by the CDS from the optical link output. The Bayesian equation, which required three real divisions per cell of the received/measured signal, was used in earlier iterations of the CDS perceptor for NGNLE [7,11,13,75,76]. A new perceptor in CDSv6 can directly extract the posterior without using the Bayesian equation to create a statistical model of the fiber optic system. The computational cost for posterior extraction is significantly reduced by this method because it does not involve any multiplications or divisions. Second, the authors of [78,79] reported an increase in the Q-factor of 1.2 dB over the system without the CDS. Third, QoT control and robustness requirements for SDOCS goal are met by offering a 1.3 dB Q-factor improvement in the presence of moderate clock recovery intolerance, which is an example of unforeseen optical channel distortions [79]. In conclusion, Table 3 compares the CDS versions 1, 3, 4, and 6 with the typical DBP method [80] and the typical AI [81].

The CDS replaces the nonlinearity compensation technique such as DBP, and from Table 4, it can be seen that the CDS requires no complex multiplications in the steady state mode, while DBP needs to use many fast Fourier transforms (FFTs) to compensate for the fiber nonlinear effects. We note that the computational cost shown in Table 3 corresponds to that of CDS only, not the cost of the entire high speed communication systems. In Table 4, *N* is the number of samples in time domain and *M* is the number of propagation steps. In CDSs v1, v3 and v4, the total computation cost of extracting the posterior is $3\times S\times F_{m}^{total,k}$ real divisions, where *S* is the number of symbols and $F_{m}^{total,k}$ is the number of cells. Previous versions of CDS [7,11,13,75,76] used this approach. However, CDS v6 requires no complex multiplications in the training phase, as well as in the steady state.

The CDS replaces nonlinearity compensation techniques such as DBP, and Table 4 shows that the CDS requires no complex multiplications in the steady state mode, while DBP needs to use numerous fast Fourier transforms (FFTs) to account for fiber nonlinear effects. It should be noted that Table 3’s computation cost only accounts for CDS and not the full cost of high-speed communication systems. The number of time domain samples (*N*) and the number of propagation steps (*M*) are shown in Table 4. In CDSs v1, v3, and v4, the computation cost for extracting the posterior is $3\times S\times F_{m}^{total,k}$ real divisions, where *S* is the number of symbols and $F_{m}^{total,k}$ is the number of cells. This strategy was used in the earlier iterations of CDS [7,11,13,75,76]. However, with CDS v6, neither the training phase nor the steady state require complex multiplications.

### 5.3. CDS for Healthcare Applications as an Example of NGNLE

The two primary CDS implementations in healthcare applications are (i) automatic diagnostic tests [71] and (ii) automatic screening procedures [74]. CDS v2 [71] and CDS v5 [74] are used to implement the automatic diagnostic test and automatic screening procedure, respectively.

#### 5.3.1. Automatic Diagnostic Test (CDS v2)

A decision tree-based CDS v2 for a diagnostic test in an intelligent e-Health home is presented in [71]. In ADMS and for cognitive decision making on a NGNLE, CDS v2 is used as an alternative to AI [71]. Through internal commands, CDS v2 can modify the perceptor’s modeling configuration by raising or lowering the decision tree level. This variant of CDS can therefore simulate an NGNLE with finite memory. Using a virtual NGNLE, the executive part, however, is unable to foresee the results of the actions without applying them on real environment.

The author of [71] applies ADMS based on CDS v2 for the diagnosis of arrhythmia. Table 5 compares the decision-making accuracy of the arrhythmia database between CDS v2 and some related options. CDS v2 performs with an accuracy of 95.4 percent, as can be seen.

#### 5.3.2. Automatic Screening Process (CDS v5)

A smart e-Health system’s screening procedure is described in [74] using CDS v5, which is based on the conflict of opinions (CoO) and a jungle of decision trees (JDT) (see Figure 16). The Perception Multiple Action Cycle (PMAC), a generalized form of the PAC, was introduced first. The PMAC can apply multiple actions simultaneously, in contrast to the typical PAC, and the perceptor can model the observables that result from the applied actions. The justification for using CoO in healthcare applications when a flawed dataset is present is given in [74]. The specific equations and algorithms for the CDS v5 screening procedure are available for the readers in [74]. Additionally, simulation results for diagnosis error and false alarm are presented in [74].

The dependency of ML-based approaches on trustworthy datasets is well known to be the natural weakness of ML-based AI in comparison to that of rule-based AI [18]. As a result, the performance of ML-based AI suffers greatly when a bad dataset is present. To lessen the impact of a flawed dataset and deliver accurate results, CDS v5 employs conflict-of-opinion (CoO) decision making. Other CDS versions (CDS v1–v4 and v6) place CDM in the perceptor; CDM based on CoO places it in the executive. Additionally, as previously mentioned, the PAC concept is expanded to include the PMAC concept in CDS v5. The NGNLE model can be extracted by the CDS v5 perceptor as a forest of decision trees [74].

**Figure 16 sensors-23-02859-f016:**
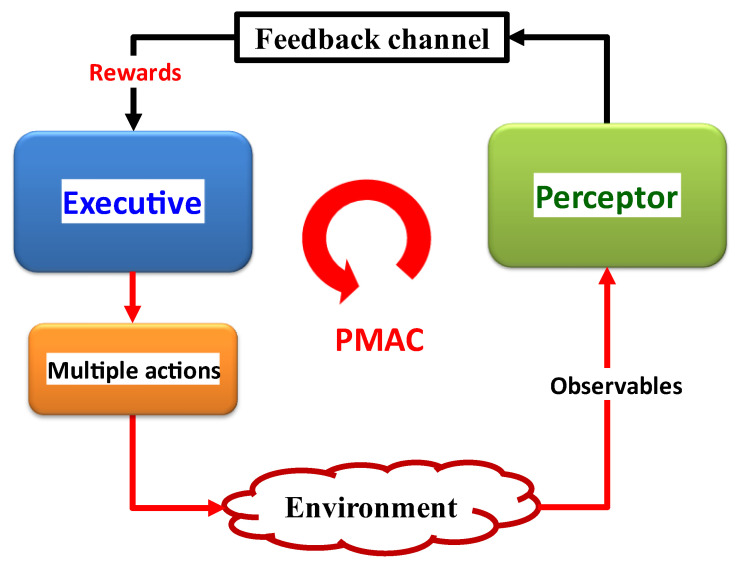
Block diagram of the perception multiple actions cycle-based cognitive dynamic system (CDS).

According to [75], the screening procedure entails a comparison of healthy and unhealthy individuals with a predetermined level of diagnosis error and an acceptable high false alarm rate. Known and unknown defective datasets are so common in the healthcare sector. For the screening procedure, some flawed datasets can be used. In order to screen for arrhythmia from a flawed dataset, a proof-of-concept case study utilizing CDS v5 is used. When they are compared to the desired diagnosis errors of 25%, 10%, 5%, and 2.5%, it is discovered that the CDS performs admirably, achieving average final diagnosis errors of 13.2%, 9.9%, 6.6%, and 4.6%, respectively. These diagnostic errors with a flawed dataset correspond to acceptable high false alarm rates of 20%, 25%, 28%, and 54.7%, respectively.

## 6. Conclusions

In conclusion, cognitive dynamic systems (CDS) have emerged as a promising alternative to traditional artificial intelligence (AI) for creating intelligent machines. The use of CDS algorithms has been explored in both linear and Gaussian environments (LGE) and non-Gaussian and nonlinear environments (NGNLE), and has yielded impressive results in various fields such as cognitive radios, cognitive radar, cyber security, self-driving cars, smart grids, and e-healthcare.

The application of CDS in LGEs has shown improved performance and accuracy, surpassing the traditional techniques in some cases. For instance, in self-driving cars and cognitive vehicular communications (CVC), the coordinated CRC (C-CRC) and C-CRC with mutual interference (C-CRC MI) algorithms outperformed fixed transmit waveform (FTW) and Q-learning techniques by up to 70%. In smart grids, CDS algorithms with cognitive risk control (CRC) can identify and control false data injection attacks (FDI) in uncertain situations such as cyberattacks. However, high computational costs have been identified as a challenge to implementing CDS on practical systems.

In NGNLEs, the results of implementing CDS in smart e-healthcare applications and software-defined optical communication systems (SDOCS) have also been promising. For example, CDS implementation in smart fiber-optic links improved the Q-factor by 7 dB and the maximum achievable data rate by 43% compared to those of other mitigation techniques. In automatic diagnostic tests and arrhythmia detection, the CDS achieved up to 95.4% accuracy, outperforming the traditional machine learning techniques.

Despite the promising results, CDS is still a new research topic that requires further development and research to fully realize its potential. The future direction of CDS research should focus on addressing the computational cost concerns, developing more powerful processors, and applying CDS to more complex NGNLE scenarios. With its potential to revolutionize the field of intelligent systems, CDS is an exciting area of research that warrants continued exploration and innovation.

## Figures and Tables

**Figure 1 sensors-23-02859-f001:**
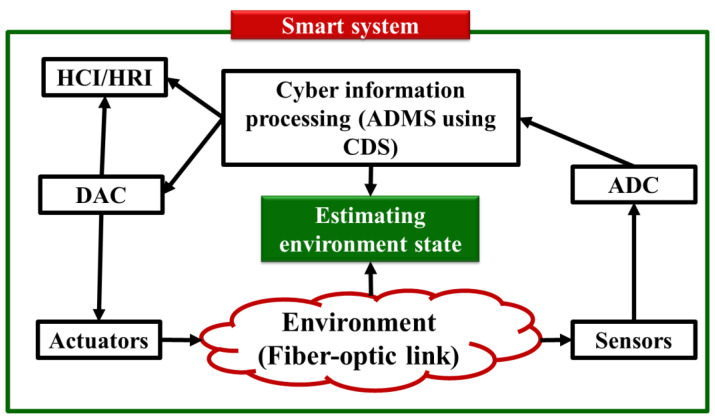
Intelligent system architecture using a cognitive dynamic system (CDS) as the cyber-physical system (CPS) and an autonomous decision-making system (ADMS). HCI: Human–computer interaction, HRI: Human–robot interaction, ADC: Analog-to-digital converter, DAC: Digital-to-analog converter.

**Figure 2 sensors-23-02859-f002:**
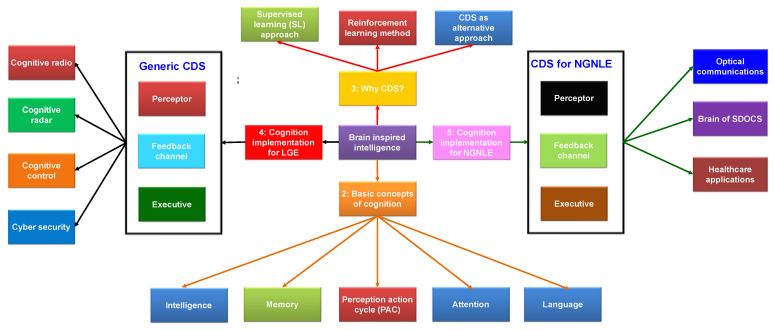
A diagram showing the association between each subsections presenting in this paper. SDOCS: software-defined optical communication system.

**Figure 3 sensors-23-02859-f003:**
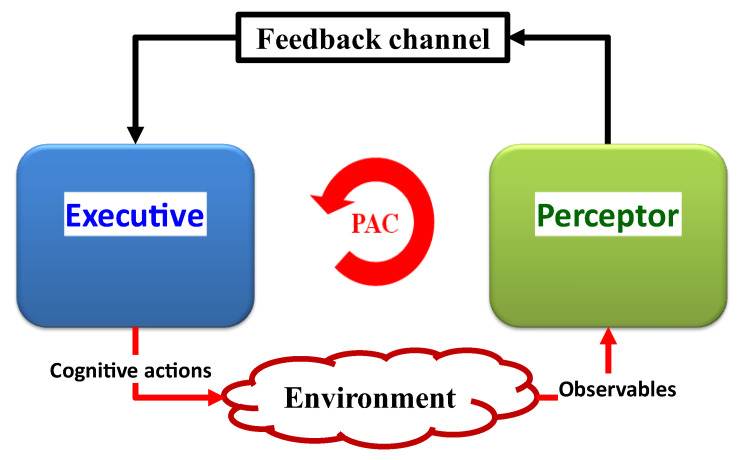
A basic cognitive dynamic system is shown in a block diagram.

**Figure 4 sensors-23-02859-f004:**
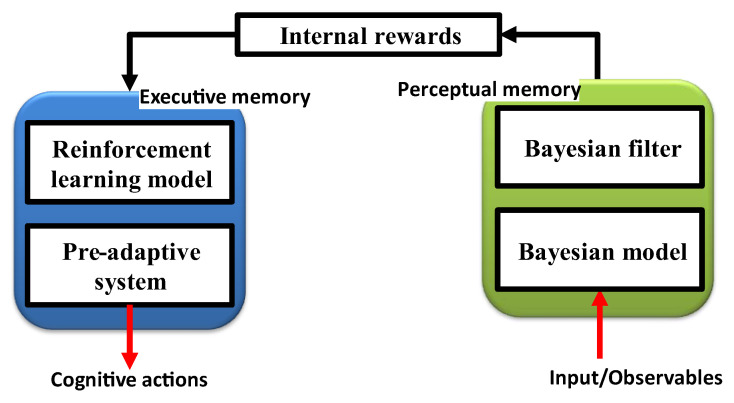
The functional brain-like block in the cognitive dynamic system that controls executive and perceptual memory. (CDS: cognitive dynamic system.)

**Figure 5 sensors-23-02859-f005:**
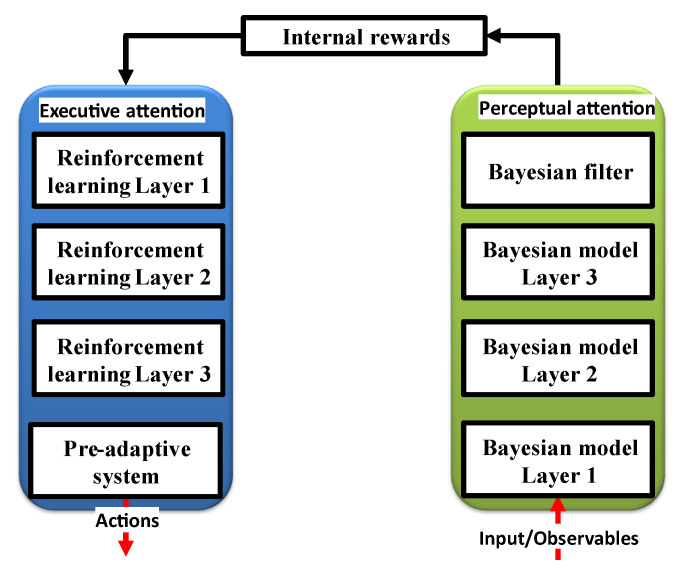
Attention/focusing in the CDS [11].

**Figure 7 sensors-23-02859-f007:**
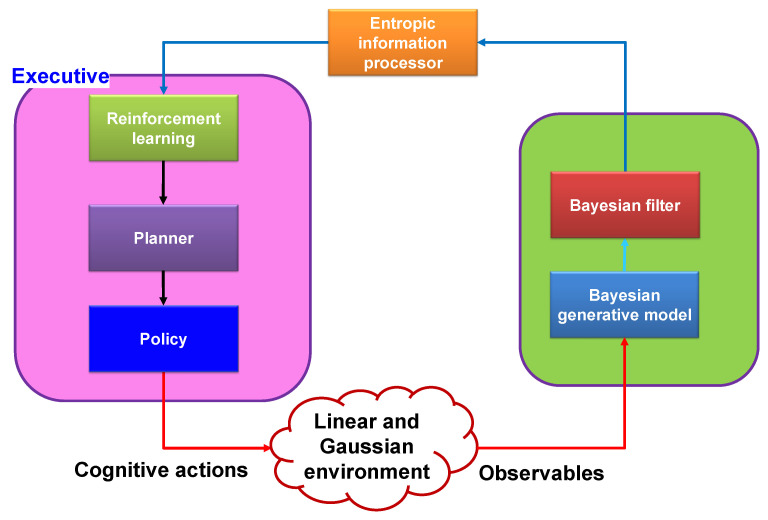
Block diagram of a typical CDS conceptually.

**Figure 8 sensors-23-02859-f008:**
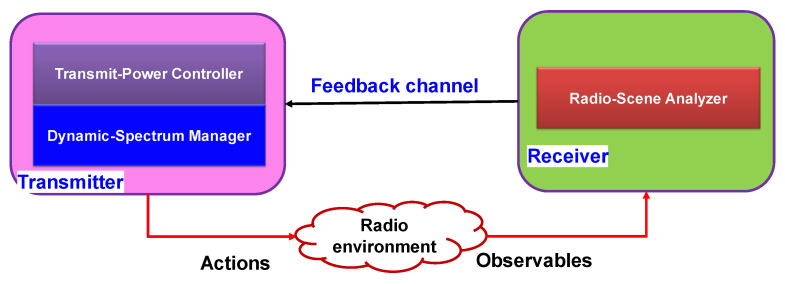
The cognitive-information-processing cycle in cognitive radio [29].

**Figure 9 sensors-23-02859-f009:**
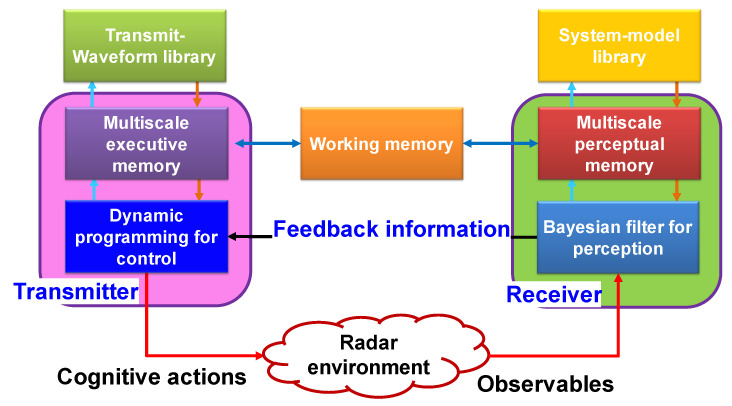
Block diagram of CR with memory.

**Figure 10 sensors-23-02859-f010:**
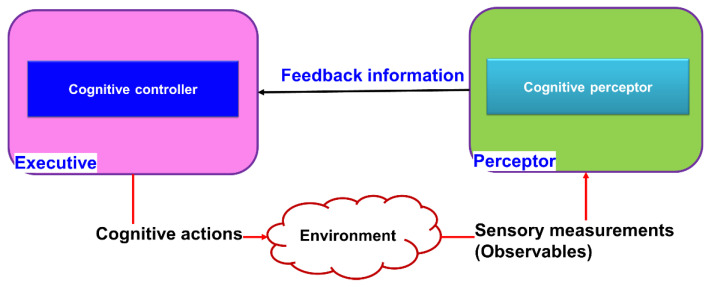
Block diagram of the overall perception–action cycle in a cognitive dynamic system.

**Figure 11 sensors-23-02859-f011:**
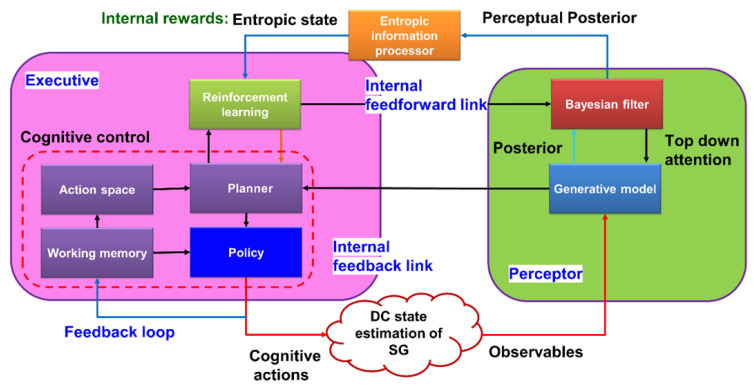
Structure of the CDS for the SG.

**Figure 12 sensors-23-02859-f012:**
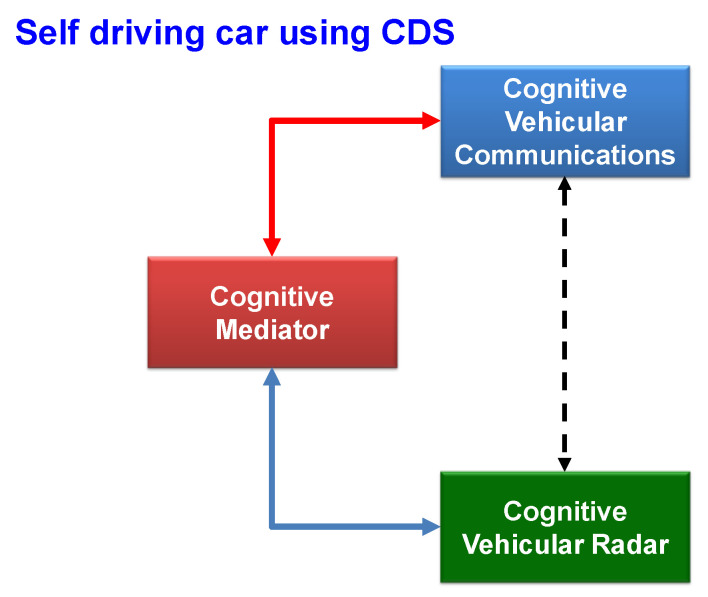
The fundamental layout of a coordinated vehicular radar and communication system.

**Figure 13 sensors-23-02859-f013:**
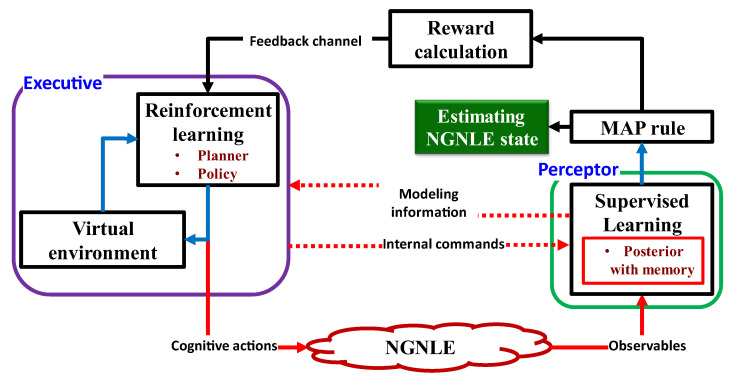
The proposed CDS’s conceptualization.

**Figure 15 sensors-23-02859-f015:**
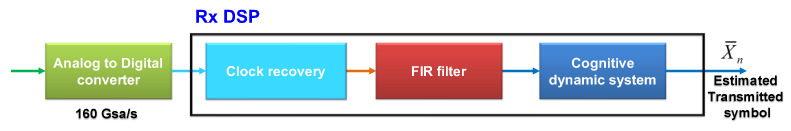
Receiver with cognitive dynamic system and OTDM digital signal processing (DSP). (CDS v6 [78]). FIR: finite impulse response and ${\bar{X}}_{n}$: Estimation of transmitted symbols.

**Table 1 sensors-23-02859-t001:** Comparison between the TAR, FAR, and CR versions [17].

	TAR	FAR	CR1	CRm	CRa
RMSE of Range (m)	29.24	2.12	0.58	0.47	0.69
RMSE of Range-rate (m/s)	51.07	13.39	4.21	3.30	3.53
Complexity	NA	NA	$O({\left(N_{g}\right)}^{L})$		

Note: Root mean square error (RMSE). NA: not applicable.

**Table 3 sensors-23-02859-t003:** Comparison of the typical AI and DBP methods and the CDS versions 1, 3, 4, and 6.

References	Technique Implemented	Q-Factor Improvement	Data Rate Enhancement	Max Bit Rate	Disturbance Resistance
CDS v1 [11,76]	Simple CDS	2.7 dB	13%	54 Gb/s	Yes
CDS v3 [7,75]	CDS with Virtual actions	3.5 dB	23%	236 Gb/s	Yes
CDS v4 [13]	Finite memory modeling + Virtual actions	7 dB	43%	280 Gb/s	Yes
CDS v6 [78]	Direct posteriori extraction for OTDM fiber optic link	1.2 dB	-	15 Tb/s	Yes
Typical nonlinearity mitigation method [82]	DBP for OFDM fiber optic link	<2 dB	-	42.8 Gb/s	No
Typical AI technique [80]	Neural Network (Steady-state mode) using 4 WDM channels	<0.6 dB	-	300 Gb/s	No

**Table 4 sensors-23-02859-t004:** In terms of computational complexity, a comparison of CDS and competing methods.

Methods in Practical Implementation	Complex Multiplication (Training Phase)	Complex Multiplication (Steady States)
CDS v1, v3, v4 [7,11,13,75,76]	$3\times S\times F_{m}^{total,k}$	None
CDS v6 [78]	None	None
DBP [82]	Not applicable	$O(MN\log_{2}N)$
Neural Network [80]	Not available	O(500×N)

**Table 5 sensors-23-02859-t005:** CDS v2 and related published works are compared.

Technique Implemented	Best Reported Accuracy (%)	Sensitivity (Diagnosis of Abnormal Rhythm Accuracy) (%)	Specificity (Normal Rhythms Accuracy) (%)
Random forest (RF) + Support vector machine (SVM) [81]	77.4	59.9	91.4
Deep learning [83]	75.8	-	-
SVM for 2 classes and 11 features [84]	86	-	-
ADMS using CDS v2 [71]	95.4	90	100

## Data Availability

Not applicable.

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
