# Peer review of "Natural Intelligence as the Brain of Intelligent Systems"

_sensors, 2023, doi:10.3390/s23052859_

Round 1

Reviewer 1 Report

The reviewer's comments are as follows:

  1. As a review paper, the abstract is hard to follow. There is too much technical jargon and other abbreviations.
  2. General block diagrams are easy to understand, but one or two could be more attractive to read. Fig 6 off-centre
  3. There is broad coverage of related technologies, but I suggest having a summary in the first section to guide the reader on various technologies and with a diagram showing the association between each sub-sections.
  4. An example of the general term could be given, such as the perceptor and controller, which are extensively used but very general terms.
  5. As a review summary, is there a trend that the CDS field is evolving? And can it be justified/illustrated by key performance metrics?

Author Response

We submit the respected reviewer response as the attached file.

Reviewer 2 Report

Reviewer #1: Comment 1: Authors work on cds and results of simulation implementation should be given in numerical form in the abstract and conclusion.

Comment 2: The work is having novel implementation review of natural intelligence algorithm scope and used method should be compared and implemented suitably.

Comment 4: The paper can be drafted in journal format and recent references  must be cited.

Comment 5: Improve the quality of figures and explain those properly.

Comment 6: There are many English and grammatical issues in the paper which needs to be rectified.

Author Response

(The authors gave the same response as above.)

Round 2

Reviewer 1 Report

Thank you for the revised manuscript, reviewer has no more comments.